# Mental Health Literacy about Personality Disorders: A Multicultural Study

**DOI:** 10.3390/bs13070605

**Published:** 2023-07-21

**Authors:** Kerim Alp Altuncu, Arianna Schiano Lomoriello, Gabriele Lo Buglio, Ludovica Martino, Asrin Yenihayat, Maria Teresa Belfiore, Tommaso Boldrini

**Affiliations:** 1Department of General Psychology, University of Padova, 35131 Padua, Italy; 2Department of Cognitive System, Denmark Technical University (DTU), 2800 Copenhagen, Denmark; arilom@dtu.dk; 3Department of Dynamic and Clinical Psychology, and Health Studies, Faculty of Medicine and Psychology, Sapienza University of Rome, 00185 Rome, Italy; 4Department of Developmental Psychology and Socialization, University of Padova, 35131 Padua, Italy; 5Department of Clinical Psychology, Psychology for Individuals, Families and Organizations, Faculty of Human Sciences, University of Bergamo, 24122 Bergamo, Italy; yenihayatasrin@gmail.com

**Keywords:** mental health literacy, personality disorders, cultural factors

## Abstract

Mental health literacy (MHL) refers to lay people’s knowledge and beliefs about the diagnosis and treatment of mental illness. The current study aimed at investigating MHL regarding personality disorders (PDs) multiculturally, comparing Turkish and Italian populations. In total, 262 participants responded to an online vignette identification task that required them to label the PDs of seven hypothetical subjects and rate various dimensions of their disorders. Narcissistic (25%), obsessive-compulsive (13%), and paranoid (12%) PDs were the most correctly labeled, while the average accuracy values for other PDs were below 0.04%. Compared to Turkish participants, Italian participants were more accurate in labeling narcissistic PD. Additionally, of the seven PDs, narcissistic PD was associated with the most happiness and success at work. Subjects with borderline and avoidant PDs were the most recognized as having psychological problems (>90%), yet their PDs were among the least correctly identified. Overall, participants from both cultures were generally successful at recognizing the presence of a mental illness, but they rarely labeled it correctly. Only limited cultural differences emerged. The present findings may inform the design of outreach programs to promote MHL regarding PDs, thereby facilitating early recognition of PDs and help-seeking behaviors for affected individuals.

## 1. Introduction

The term “mental health literacy” (MHL) refers to lay people’s knowledge and beliefs about mental disorders, with respect to their recognition, management, and prevention. Specifically, MHL describes the ability to recognize specific disorders; knowledge of how to acquire mental health information; knowledge of risk factors and causes as well as the forms of professional help that are available; and attitudes that promote the recognition of PDs and appropriate help-seeking behaviors [1]. Accordingly, MHL is considered a core domain of mental health [2].

Because MHL has been shown to facilitate help-seeking behavior [3] and improve community support for mental health issues [4], it has received increasing attention as a strategy for promoting the early identification of mental disorders and the development and implementation of effective treatment interventions [5]. Of note, research has shown that MHL can be improved [6]. In support of this aim, MHL measures evaluating knowledge about and attitudes towards mental disorders, as well as help-seeking behavior, have been developed [7], and programs/interventions to promote MHL among youths in school settings have been designed, applied, and empirically tested [8,9].

MHL research may be particularly relevant to personality disorders (PDs), as a lack of PD recognition and help-seeking behavior is one of the main barriers to PD treatment. Because PDs are trait-like and egosyntonic conditions, individuals with PDs may be generally unaware of their problematic patterns and only seek treatment following the urging of others. Although some pursue treatment through self-referral, they often do so for more circumscribed and state-like distress experiences, such as those associated with mood, anxiety, and substance-use disorders [10]. As a result, many PDs remain undetected, or their diagnosis is delayed. Unfortunately, “delayed intervention” is associated with suboptimal outcomes, including functional impairment, disability, and therapeutic nihilism [11,12]. 

Previous studies on PD literacy have shown that, although a large proportion of lay people are able to perceive when a psychological problem is present, most are unable to assign the correct diagnostic label. For example, studies using a diagnostic vignette approach have found that 7 out of 10 PDs are correctly labeled less than 7% of the time [13,14]. In contrast, other research has shown that depression and schizophrenia are correctly labeled more than 60% of the time [15], highlighting that PD literacy is much lower than that of other mental disorders. Thus, there is a need for greater mental health awareness and psychoeducation in order to increase help-seeking behavior among afflicted individuals. 

To the best of our knowledge, most prior research on PD literacy has been conducted in English-speaking populations, and predominantly in the UK [13,14,15,16,17]. Because different cultures may have different views on how various PDs manifest, current evidence may be “culture bound” and not easily generalized. Importantly, the impact of key contextual factors (e.g., culture, ethnicity, geographic location, education, health system) on MHL is unknown [7]. Thus, multicultural research is needed to establish the feasibility and potential adaptation of programs/interventions to promote MHL within specific cultural contexts. Furthermore, multicultural differences may provide a starting point for deeper investigations into the impact of local phenomena and cultural psychological constructs [18]. 

The present study aimed at investigating PD MHL multiculturally, comparing Turkish and Italian lay people, specifically. Cultural differences between Italy and Turkey were compared using Hofstede’s dimension, which is based on six dimensions: power distance (i.e., attitudes towards inequality), individualism versus collectivism (i.e., interdependence within society), masculinity versus femininity (i.e., emphasis on competition vs. quality of life), uncertainty avoidance (i.e., attitudes towards future uncertainty), long-term orientation (i.e., emphasis on tradition vs. pragmatism, when dealing with challenges), and indulgence versus restraint (i.e., the extent to which desires and impulses are controlled) [19,20]. A previous meta-analysis of 180 studies conducted across 38 countries found support for these dimensions [21]. 

In Turkey, power is centralized, and there is greater acceptance of hierarchical relationships in society (i.e., high power distance); in contrast, Italian society is more egalitarian (i.e., low power distance). In Italy, a “me”-centered culture is dominant (i.e., high individualism), and there is a high orientation towards success (i.e., high masculinity). In Turkey, however, a “we”-centered culture is dominant (i.e., low individualism) and people belong to in-groups; furthermore, communication is less direct and the softer aspects of culture (e.g., leveling with others, consensus, sympathy for the underdog) are valued, indicating low masculinity. Neither Turkish nor Italian lay people are comfortable in ambiguous situations (i.e., high uncertainty avoidance). However, they express this discomfort differently. Turkish people make use of many rituals that, to foreigners, may seem religious (i.e., with many references to “Allah”), but are simply traditional social patterns used in specific situations to ease tension. Italian people value formality and have a penal and civil code complicated by clauses and codicils. Furthermore, Italian culture shows an ability to adapt traditions to suit changing conditions, as well as a strong propensity to save and invest (i.e., high long-term orientation); it is also a culture of restraint (i.e., low indulgence). Restrained societies tend to de-prioritize leisure time and control desires. In comparison, Turkey’s intermediate scores for long-term orientation and indulgence do not allow a dominant cultural preference to be inferred [22,23]. 

The abovementioned differences suggest that the Turkish and Italian cultures may differ in their perception and identification of individual patterns of inner experience and behavior (i.e., personality traits, PDs) as deviations from cultural expectations. Accordingly, the present study aimed at investigating Italian and Turkish participants’ ability to label various PDs (i.e., correctly label diagnoses) and identify pathology/normality, as well as their perceptions of three domains of living with a PD (i.e., happiness, success at work, relationship satisfaction); it also explored potential cultural predictors of PD literacy (e.g., sociodemographic factors, familiarity with mental illness). 

## 2. Materials and Methods

### 2.1. Study Design

An online survey designed in Qualtrics was administered between 25 September and 31 October 2022. Participants were recruited using snowball sampling. Specifically, the open link to the survey was shared with the authors’ personal contacts, as well as on social media platforms. Within the survey, participants were offered two language options (i.e., Turkish or Italian). Participants were also told within the questionnaire that their task would be to read vignettes portraying people in their daily lives in different contexts, and to subsequently label the subjects’ PDs and rate them for various characteristics. Participation in the research was voluntary, and no incentives were provided. 

All participants provided informed consent by agreeing to the data protection declaration prior to starting the survey. The principles outlined in the Declaration of Helsinki were followed, ensuring anonymous participation through the administration of the informed consent format of the ethics committee of the University of Padua (GDPR EU 2016, pd. 196/03).

### 2.2. Mental Health Literacy Measure 

The study used a diagnostic vignette approach to measure MHL. While this approach has been widely used as a measure of mental health knowledge [1,7,13,14,15,16,17], it has some limitations, including (1) uncertainty in regard to the clarity with which the vignettes describe all the salient behaviors associated with each disorder [7,24], (2) the potential for subtle wording issues to lead to particular results [14], and (3) concern over participants’ ability to discriminate between “normal” and “ill” subjects [25]. The latter limitation may be less relevant to PDs, given increasing research on PDs through a dimensional model [26]. Despite these limitations, we opted for the diagnostic vignette approach, as it has been tested and validated in previous studies [13,14,15,16,17] and is feasible for survey studies. 

Specifically, we translated, updated, and adapted the questionnaire developed by Furnham et al. [13,14], which consists of 10 vignettes describing three cluster A (i.e., schizotypal, paranoid, schizoid), four cluster B (i.e., antisocial, borderline, histrionic, narcissistic), and three cluster C (i.e., avoidant, dependent, obsessive-compulsive) PDs. Each vignette was accompanied by the same five questions regarding the adjustment of the vignette subject to living with his/her PD. The final question asked respondents to determine whether the subject had a psychological problem (i.e., pathology). If participants responded affirmatively, they were asked to apply a label to the psychological problem, in response to the open-ended question: “If so, what is it?” Participants’ qualitative responses to this question were coded into categories, in order to maximize the identification of response patterns. Participants who answered “No” to the question, “Do you think that, in any sense, they have a psychological problem?” were not asked the labeling question. Instead, their (lack of) response was coded as “None”. Finally, participants were asked whether they had ever received treatment for mental illness and whether they knew of someone who had received treatment for mental illness (as potential predictors of MHL).

The original version of the questionnaire was translated into both languages (i.e., Italian, Turkish) by two members of the research team who were fluent in both English and the target language, and then back translated into English by native English speakers who were also fluent in Italian or Turkish. After the instrument was administered to colleagues, the need to change the names in the vignettes to fit the local culture became apparent (e.g., Barry was translated into Berkay and Bruno for the Turkish and Italian versions, respectively). Moreover, the vignettes were slightly modified to align with the new DSM-5-TR [27] criteria. Finally, to make the questionnaire less time-consuming to complete and to improve compliance, three PDs were removed from the final version (i.e., dependent, schizoid, histrionic). Schizoid PD was removed due to its overlapping characteristics with schizotypal PD, and dependent and histrionic PDs were removed because they are less prevalent than the other PDs in their respective clusters [28]. To prevent potential gender bias, vignette subjects were assigned a gender at random.

An example vignette with corresponding questions is presented in Appendix A.

### 2.3. Participants

In total, 262 participants took part in the study. Of these, 128 were male (48.9%). Participants ranged in age from 18 to 78 years (*M* = 44.18, *SD* = 16.76). Additionally, there were more Turkish (*n* = 181, 69.1%) than Italian participants (*n* = 81, 30.9%). Table 1 reports the full characteristics of the sample. 

### 2.4. Inclusion Criteria for Labels and Survey Entries

Respondents were required to answer all questions associated with at least four vignettes in order to be considered for the data analysis. For all PDs, only the name of the PD (e.g., narcissistic, borderline, obsessive-compulsive) was considered acceptable as a correct label, with no further requirement for the inclusion of “personality disorder”. For narcissistic PD, both “narcissistic” and “narcissism” were accepted as correct labels; for obsessive-compulsive PD, “obsessive”, “OCD”, and “obsessive-compulsive” were accepted as correct. Labels could include more than one answer.

### 2.5. Statistical Analysis

Generalized linear mixed-effect models (GLMMs) were used to test whether different ethnicities performed differently on the vignette identification task. To measure participant accuracy in the labeling of PDs and the identification of pathology/normality, we used a binomial family distribution, as the outcome variables (i.e., correct vs. incorrect) were dichotomous. In contrast, we used a normal distribution for the three GLMMs run for “happiness”, “relationship satisfaction”, and “success at work”. The fixed effects for all three models included ethnicity (i.e., Italian, Turkish), PDs (i.e., antisocial PD [APD], avoidant PD [AVO], borderline PD [BPD], narcissistic PD [NPD], obsessive-compulsive PD [OCDP], paranoid PD [PPD], schizotypal PD [STPD]), and the interaction between ethnicity and PDs (i.e., the full model structure for all outcomes in Wilkinson notation was as follows: dependent variable ∼ ethnicity × PDs + [1| ID]). The structure of the random effect included participants as random intercepts, thus adjusting for individual differences in the dependent variable (i.e., labeling accuracy, perceived pathology). To test the predictors of literacy, we ran a GLMM including age, gender, education, profession, previous treatment for mental illness, and knowledge of someone who had received treatment for mental illness as fixed effects, and the subject as the random intercept. We implemented the same approach as that used in previous studies (e.g., with neural data: [29,30,31]; with behavioral data: [32,33,34]). All analyses were performed using the software R (2.13), applying the “lmer” function from the lme4 package ([35]). Significance levels for fixed effects were computed using the “anova” function in the lmerTest package, which employs Satterthwaite’s approximation for degrees of freedom. Post hoc comparisons were computed using the PHIA package (i.e., post hoc interaction analysis), corrected for multiple comparisons using the false discovery rate ([36]). Confidence intervals (set at 95%) were defined only for paired post hoc comparisons, with reference to the difference between means (i.e., *M_diff_*, as suggested by [37]).

## 3. Results

### 3.1. Labeling PDs 

With regard to participants’ PD labeling accuracy, the results showed a main effect of PDs, χ^2^ (6, *N* = 254) = 171.95, *p* < 0.001, with a medium to large association between PDs and participant responses (η^2^_p_ = 0.67). Specifically, participants were more accurate in recognizing NPD (*M_accuracy_* = 0.25), OCDP (*M_accuracy_* = 0.13), and PPD (*M_accuracy_* = 0.12), compared to the other PDs (all *M_accuracy_* < 0.004). An interaction was also found between PDs and ethnicity χ^2^ (6, *N* = 254) = 36.164, *p* < 0.001. The planned comparisons indicated that Turkish participants were significantly less accurate in identifying NPD relative to Italian participants, χ^2^ (1, *N* = 254) = 40.42, *p* < 0.001, *M_diff_* = 0.23 (see Figure 1).

Appendix A shows the ranking of participant labels (grouped into categories, in order to reduce variability). Most commonly, STPD was labeled “schizophrenia” (8.4%) and “denial of reality” (8.8%); PPD was labeled “trust issues” (18.3%) and “skepticism/skeptical” (12.2%); APD was labeled “lying/mythomania” (6.9%); BPD was labeled “bipolar” (12.3%) and “unbalanced/unhappy” (7.7%); NPD was labeled “selfish/egoist” (11.1%) and “approval/attention seeking” (5%); AVO was labeled “insecurity/low self-esteem” (35.11%); and OCPD was labeled “control freak/control issues” (9.2%). 

### 3.2. Perceived Pathology 

With regard to the accuracy with which participants identified pathology/normality, the results showed a main effect of PDs, χ^2^ (6, *N =* 254) = 190.72, *p* < 0.001, with a medium to large association between PDs and participant responses (η^2^_p_ = 0.43). Specifically, participants demonstrated an above-chance level of accuracy in their identification of pathology (*t*[253] = 980.35; *p* < 0.001; *M* = 0.76) for all PDs, with the greatest accuracy achieved for BPD (*M_accuracy_* = 0.94) and AVO (*M_accuracy_* = 0.91). An interaction was also found between PDs and ethnicity, χ^2^ (6, *N* = 254) = 15.491, *p* = 0.02. The planned comparisons indicated that Turkish participants were significantly more accurate than Italian participants in identifying pathology for OCDP, χ^2^ (1, *N* = 254) = 6.94, *p* = 0.03, *M_diff_* = 0.09, and PPD, χ^2^ (1, *N* = 254) = 7.67, *p* = 0.01, *M_diff_* = 0.23, only (see Figure 2).

### 3.3. Dimensional Ratings

Three separate GLMMs were run for the different PD dimensional ratings. For all ratings, a significant main effect of PDs was found (happiness: *F*[6, 179] = 32.79, *p* < 0.001; success at work: *F*[6, 156] = 28.15, *p* < 0.001; relationship satisfaction: *F*[6, 13.66] = 2.65, *p* = 0.014). For NPD only, happiness and success at work were rated significantly higher. Hence, participants associated NPD with greater happiness and success at work than other PDs. Regarding relationship satisfaction, ratings were higher for NPD, OCDP, and PPD compared to the other PDs (see Figure 3).

For success at work only, a significant interaction emerged between PDs and ethnicity, χ^2^ (6, 13.12) = 2.36, *p* = 0.028. The planned comparisons indicated that Italian participants associated PPD with greater success at work compared to Turkish participants, χ^2^ (1, *N* = 254) = 10.69, *p* = 0.007, *M_diff_* = 0.41 (see Figure 4).

### 3.4. Predictors of Literacy

Among the model parameters (i.e., age, gender, education, profession, previous treatment for mental illness, knowledge of someone who had received treatment for mental illness), only participant age had an effect on the accuracy of PD labeling (χ^2^ [6, 13.12] = 44.55, *p* < 0.001). Figure 5 shows the inverse relationship between the two parameters: as age increased, accuracy decreased.

## 4. Discussion

The present study aimed at investigating MHL about PD on a multicultural level, comparing the Turkish and Italian populations. The results showed that participants from both cultures tended to recognize pathology (*M* = 76%), but typically failed to label it correctly. The most correctly labeled PDs were NPD (25%), OCPD (13%), and PPD (12%), while the labeling accuracy for other PDs remained below 0.04%. This finding is partially in line with the results of Furnham and Wincenslaus, showing that OCPD (15.2%) and NPD (11.7%) were more frequently identified than other PDs. One possible explanation for this is that terms such as “obsessive-compulsive”, “paranoid”, and “narcissism” are frequently used in the media, and lay people may be widely exposed to them [13]. Moreover, OCPD has the highest prevalence in the general population [28], and the prevalence of NPD may have increased in recent decades due to sociocultural factors [38]. These aspects may influence knowledge about the two disorders among the general population. Finally, in the present study, younger people were better able to identify PDs compared to older people—possibly due to the significant amount of PD information that circulates on social media, which younger people tend to use to a greater extent [39].

While BPD was the most frequently identified as pathology among all of the PDs, it was also one of the least likely to be correctly labeled. The characteristics of BPD include emotional instability, impulses of anger, promiscuous sexual relationships, self-harming behaviors, and substance abuse [27], and it is possible that these externalizing and dramatic symptoms may have made the presence of psychological problems easier to identify. However, it seems that participants struggled to correctly identify the gestaltic explanation for the co-occurrence of these symptoms (i.e., BPD). This mirrors a pattern observed in clinical settings, wherein mental health professionals commonly focus on “isolated” symptoms or comorbidities (i.e., 85% of BPD patients have at least one additional non-PD [40]) that saturate the clinical picture, rather than the patient’s full set of personality features. As a result, individuals with BPD symptoms often access evidence-based treatment for BPD only when they reach a state of chronic psychosocial dysfunction and suffer from frequent mental health crises [41].

Additionally, the present study found no differences in PD labeling accuracy between Turkish and Italian participants, except in the case of NPD, which was better recognized by Italian participants. It may be the case that some cultural and contextual characteristics of Italian society increased Italian participants’ recognition of narcissism. For instance, according to Hofstede’s theory ([22,23]; see “Introduction”), Italian society tends to be more egalitarian (i.e., characterized by lower power distance) than Turkish society. This may have led Italian participants in the present study to perceive the subject’s sense of entitlement in the NPD vignette as more culturally deviant, potentially increasing their recognition of this disorder. On the other hand, Turkish society is considered more collectivistic than Italian society, and this may have driven the results in the opposite direction. Ultimately, cultural differences are complex and multifaceted, and deep sociological explanations are beyond the scope of this study.

Conversely, some cultural differences emerged for the dimensional ratings attached to certain PDs. Specifically, compared to Turkish participants, Italian participants associated PPD with greater success at work. A possible explanation for this finding is that distrust of the other may be seen as less adaptive in predominantly collectivistic cultures, such as that of Turkey ([22,23]). Turkish participants were also significantly more accurate than Italian participants in their identification of pathology in the context of both PPD and OCDP. With regard to OCDP, the vignette subject was overly focused on work, to the detriment of his married life. Thus, the difference may be explained by the fact that Turkish culture tends to emphasize feminine qualities (e.g., quality of life, harmony), whereas Italian culture tends to emphasize masculine qualities (e.g., ambition, drive) ([22,23]).

The present results suggest that culture may influence individuals, families, and groups in unique ways, establishing culture-specific norms for behavior and personality traits [18]. Research on PDs that situates the individual in a sociocultural context may contribute to improving cultural competence among clinicians [42], helping them to assess and treat patients in a more contextually grounded manner (i.e., considering not only the dynamic interactions between personality traits, developmental histories of adversity, and social contexts, but also the process of recovery) [42,43]. Of note, a recent position statement by the Canadian Psychiatric Association [42] underlined the importance of integrating culture into the psychiatry curriculum. Indeed, culture may impact not only the mental health, symptomatology, and clinical presentation of patients, but also clinicians’ ability to correctly identify mental disorders and determine appropriate interventions. Further research focusing on the inter-relationship between culture and mental health is required to address cultural differences in clinical services.

In the present study, both Turkish and Italian participants rated NPD highest for happiness and success at work and, together with OCD and PPD, also relationship satisfaction. This suggests that lay people may associate narcissistic traits with success and well-being, in line with modern society’s increased focus on individualism, fame, and celebrity [38,44]. In this vein, scholars have argued that narcissism may have transformed from a niche to a mass phenomenon [45], and it should, therefore, be contextualized both historically and socioculturally [38].

Importantly, some limitations of the present study must be acknowledged. First, the study produced only partial evidence on PD literacy, given that three PDs were removed from the questionnaire in order to improve its accessibility. Second, although the diagnostic vignette approach is widely used as a measure of MHL, a study comparing diagnostic vignettes with non-diagnostic vignettes showed that, in the former approach, participants were unable to discriminate between “normal” and “ill” categories [25]. Thus, further research is needed to establish the validity of the diagnostic vignette as a measure of mental health knowledge [7]. Third, the snowballing technique that was used to recruit participants may have led to some selection biases. However, we are confident that the way in which the survey was initially presented to participants (i.e., as a set of vignettes portraying subjects in their daily lives, in different contexts) was sufficiently general to control for this potential source of bias. Finally, as the investigated cultural contexts were limited to Italy and Turkey, replications in other cultural and geographic contexts are needed.

## 5. Conclusions

The present study was the first to investigate multicultural differences in MHL with regard to PDs among lay people. The results showed relatively limited cultural differences. Specifically, cultural factors were found to influence the correct labeling of NPD (but not other PDs) and the identification of pathology (irrespective of the ability to correctly label the PD). Thus, MHL about PDs requires significant improvement across cultures. The present results corroborate the findings of previous studies conducted in English-speaking populations, while highlighting the need for MHL training. Indeed, good mental health can be defined by different core domains—including MHL [2]—which can be targeted and potentially improved through universal and selective interventions [46].

Future research should seek to explore whether participants from different countries respond differently to interventions aimed at increasing MHL about PDs. Further multicultural studies adopting a longitudinal research design may help with culturally informed programs aimed at enhancing MHL and, consequently, promoting mental health (e.g., by improving the capacity to recognize PDs, which in turn may foster help-seeking behavior). Of note, future research may inform clinicians and health care systems and, as a result, promote culturally sensitive clinical practice [42].

## Figures and Tables

**Figure 1 behavsci-13-00605-f001:**
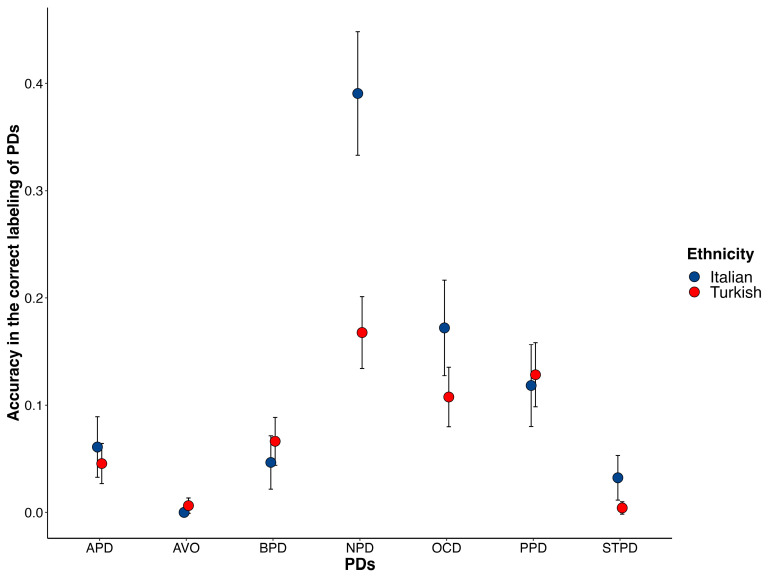
Average accuracy in the vignette identification task for each PD in both groups (i.e., Italian, Turkish). Bars represent 95% confidence intervals.

**Figure 2 behavsci-13-00605-f002:**
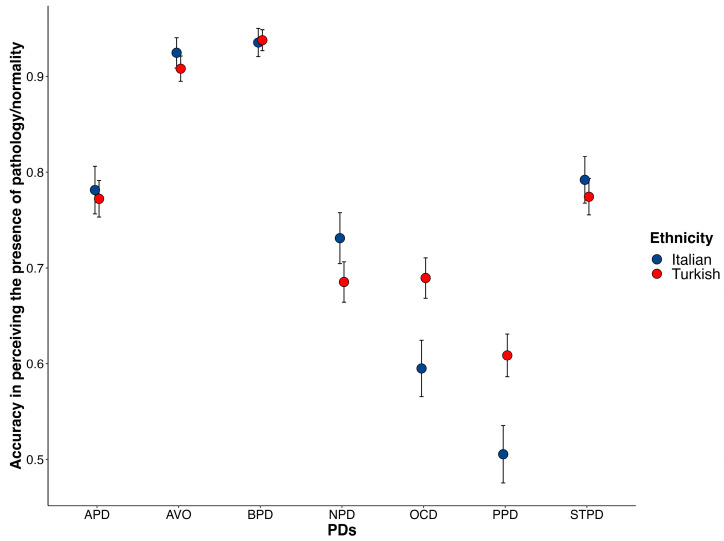
Average accuracy in identifying pathology/normality for each PD in both groups (i.e., Italian, Turkish). Bars represent 95% confidence intervals.

**Figure 3 behavsci-13-00605-f003:**
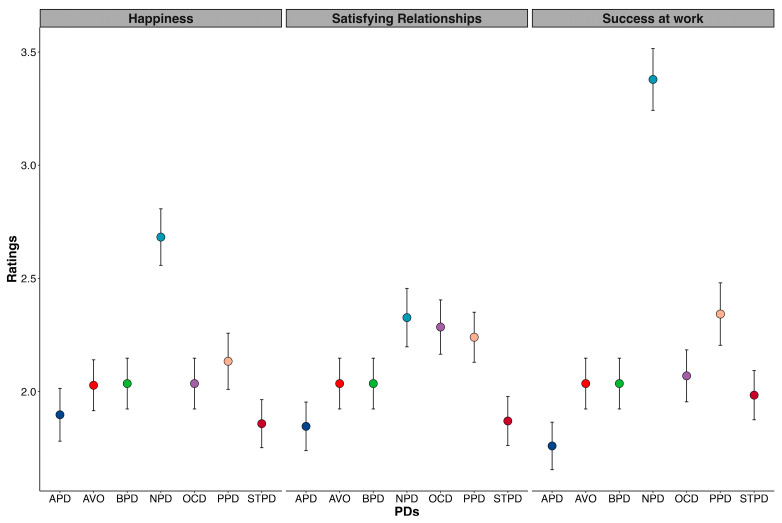
Dimensional ratings (i.e., happiness, relationship satisfaction, success at work) for each PD. Bars represent 95% confidence intervals.

**Figure 4 behavsci-13-00605-f004:**
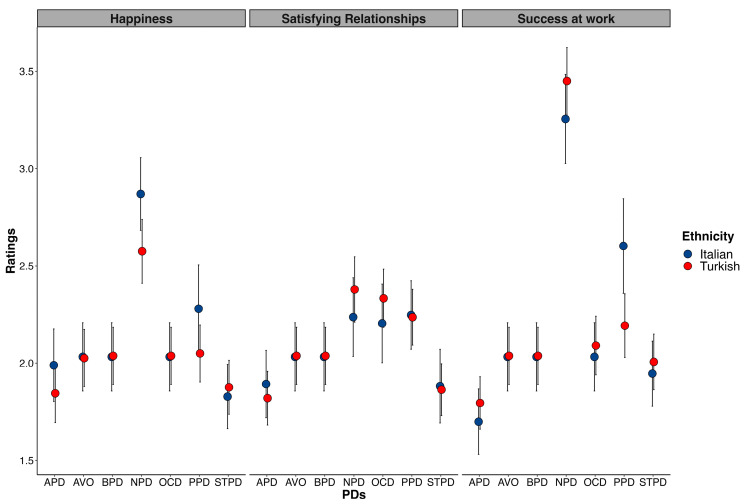
Italian and Turkish participants’ dimensional ratings (i.e., happiness, relationship satisfaction, success at work) for each PD. Bars represent 95% confidence intervals.

**Figure 5 behavsci-13-00605-f005:**
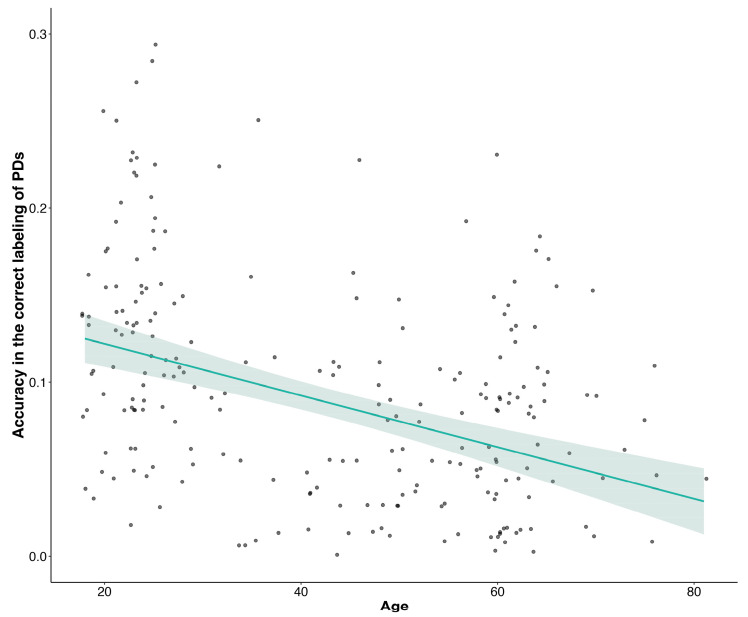
Mean accuracy in the vignette identification task as a function of participant age. Dots represent mean response values.

**Table 1 behavsci-13-00605-t001:** Sample characteristics.

	Total (N = 254)	Turkish (N = 93)	Italian (N = 161)
**Gender**			
Female	138 (54%)	66 (71%)	72 (45%)
Male	116 (46%)	27 (29%)	89 (55%)
**Age (years)**			
Mean (SD)	43 (±18)	37 (±18)	47 (±16)
**Education**			
BA	121 (48%)	26 (28%)	95 (59%)
HS	75 (30%)	33 (35%)	42 (26%)
MA	42 (17%)	24 (26%)	18 (11%)
Missing data	10 (4%)	7 (8%)	3 (2%)
PHD	6 (2%)	3 (3%)	3 (2%)
**People who underwent mental treatment**			
no	218 (86%)	72 (77%)	146 (91%)
yes	32 (13%)	19 (20%)	13 (8%)
Missing	4 (1.6%)	2 (2.2%)	2 (1.2%)
**People who know someone who underwent mental treatment**			
Yes	128 (50%)	60 (65%)	68 (42%)
No	126 (50%)	33 (35%)	93 (58%)
**Profession**			
Health Related	20 (8%)	2 (2%)	18 (11%)
Mental Health Related	3 (1%)	2 (2%)	1 (1%)
Other	3 (1%)	3 (3%)	0 (0%)
Law Related	30 (12%)	0 (0%)	30 (19%)
Student	55 (22%)	35 (38%)	20 (12%)
Engineering/Architecture	33 (13%)	9 (10%)	24 (15%)
Skilled Labourer	6 (2%)	0 (0%)	6 (4%)
Work within Education and Social Science	22 (9%)	4 (4%)	18 (11%)
Publich Servant and Clerks	23 (9%)	9 (10%)	14 (9%)
Work within Tourism, art, and sport	29 (11%)	11 (12%)	18 (11%)
Work within Finance, Business, and Commuinication	1 (0%)	1 (1%)	0 (0%)
Missing	29 (11.4%)	17 (18.3%)	12 (7.5%)

## Data Availability

All data are contained within the article and Appendix A.

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
