# Peer review of "Mental Health Literacy about Personality Disorders: A Multicultural Study"

_behavsci, 2023, doi:10.3390/bs13070605_

Round 1

Reviewer 1 Report

The authors' study on the mental health literacy of 262 study participants in Italy and Turkey regarding personality disorders makes an important contribution to the literature, as it has the potential to assist in designing effective mental health interventions. However, there are some areas that require improvement, which I would like to address:

Mauchly's test for sphericity: In the Methods section, the authors should explain their use of Mauchly's test for sphericity and provide the rationale for its application. This statistical test is used to evaluate the assumption of sphericity in the data and it is important that the authors justify its use in their study.

Tables 3, 4, & 5: In Tables 3, 4, & 5, the authors should include the meanings of the superscripts a, b, and c in the table footnotes to facilitate readers' comprehension of the statistical results.

Conclusions: The authors' conclusions should be revised to provide a concise summary of the research question, pertinent findings, implications, and future directions. While the authors' claim that their study is the first of its kind in Italy and Turkey may be inaccurate, they should highlight the strengths of their study and how it advances existing literature. Furthermore, it is not conventional to include in-text citations in the conclusions section, so the authors should consider removing them.

Reviewer 2 Report

We were very pleased to read the article entitled "Mental Health Literacy on Personality Disorders: a Multicultural Study". It is an interesting article that looks at people's ability to label personality disorders based on the clinical vignettes provided. The article is of good quality and does not present any major difficulties. We have highlighted a few elements that would improve the quality of the article. The first is the statistic analyses. The whole article is modelled as if the distribution of the data was parametric. However, no test of normality is performed or mentioned. It would be better to start with that.

The second point seems more important to us. In the title the article refers to the multicultural issue. This is very well presented in the introduction but also very well discussed. However, almost all the results do not refer to this. Only paragraph 3.2 mentions the multicultural dimension. This paragraph consists of 5 lines and refers to a table in the annex. This shows to what extent this theme is not central in the results section. More results related to this aspect should be presented. Are there differences according to gender, age etc.?

A problem with paragraph 3.3 is that it refers to a figure 2, which is not present in the article nor in the supplementary material. If this figure is to be included in the supplementary material, it should be indicated. In the submitted document, the supplementary material only contains a figure 1.

Regarding the discussion, we invite the authors to take a cross-Cultural approach (e.g. Kirmayer LJ, Fung K, Rousseau C, Lo HT, Menzies P, Guzder J, Ganesan S, Andermann L, McKenzie K. Guidelines for Training in Cultural Psychiatry. Can J Psychiatry. 2021 Feb;66(2):195-246. doi: 10.1177/0706743720907505. Epub 2020 Apr 29. PMID: 32345034; PMCID: PMC7918872.)

Especially since the authors refer to a specialist in this field (reference 24; note that there is a spelling mistake in the writing of Kirmayer's name, there is no (b) at the end).

We would like to congratulate the authors for this work.

Reviewer 3 Report

Thank you for the opportunity to review this paper, which I found very interesting to read. This is a well-designed study of the relationships between mental health literacy and personality disorders. But I don’t think the paper is ready for publication in its current form and it should be improved adequately. Some suggestions for revisions are provide below.

Ø  Introduction part

1.         This article is a research done in different cultural background and the purpose of the research is to verify whether the previous research results carried out in the English-speaking population are culturally universal.

2.         This article is a study done in different cultural background and the purpose of the study is to verify the cultural universality of the previous research results carried out in the English-speaking population, but the author did not introduce the differences between Turkish culture and Italian culture. The author should introduce the two cultures and the understanding of personality disorders based on these two cultural backgrounds. Then based on these two cultural differences, the author can reasonably make the hypothesis of this study.

3.         The article did not present a hypothesis for this study. The article should put forward the hypothesis of this study based on the previous research results and the analysis of the differences between the two cultures.

Ø  Materials and Methods

4.         No introduction of the measurement tool(a diagnostic vignette approach)used in this study and why it was chosen. The article should detail why this measurement was used and its advantages and disadvantages.

5.         Snowball sampling method was used to select participants. This method may lead to bias in the sample as a whole, thereby affecting the results. The procedures for the recruitment of participants are written very briefly, and can be written in more detail.

6.         The gender in the vignette example is male. Will this cause better performance of  identification and accurate labeling of females than males in the results? That is, the male and female protagonists of the experimental materials should be more gender-balanced, but the article does not detail elaborate.

Ø  Discussion

7.         The article does not explain the significantly differences in the rating of "happiness" and "success at work" for the various personality disorder characters, but no significantly differences in the rating of “satisfying relationships”.

8.         The articel does not explain why people who have treatment for a psychological illness and know someone who had treatment are better at identifying and labeling of certain personality disorders.

9.         It does not discuss in depth why the Italian participants was better at identifying narcissistic personality than the Turkish participants. The article stays on the superficial discussion. For example, the religious beliefs of the two countries are different. One believes in Catholicism and the other believes in Islam. Could this be the reason for the difference?

10.      It doesn't say anything about the significance of this study, that is, the doing of this study and what the results of this study mean for the field of study.

Round 2

Reviewer 1 Report

The authors have addressed all my concerns, and the manuscript is much improved and suitable for publication. Thank you for the opportunity to review this valuable work.